# The Effects of Peripubertal THC Exposure in Neurodevelopmental Rat Models of Psychopathology

**DOI:** 10.3390/ijms24043907

**Published:** 2023-02-15

**Authors:** Martina Di Bartolomeo, Tibor Stark, Serena Di Martino, Fabio Arturo Iannotti, Jana Ruda-Kucerova, Giovanni Luca Romano, Martin Kuchar, Samuele Laudani, Petr Palivec, Fabiana Piscitelli, Carsten T. Wotjak, Claudio Bucolo, Filippo Drago, Vincenzo Di Marzo, Claudio D’Addario, Vincenzo Micale

**Affiliations:** 1Department of Bioscience and Technology for Food, Agriculture and Environment, University of Teramo, 64100 Teramo, Italy; 2Scientific Core Unit Neuroimaging, Max Planck Institute of Psychiatry, 80804 Munich, Germany; 3Department of Pharmacology, Faculty of Medicine, Masaryk University, 62500 Brno, Czech Republic; 4Department of Biomedical and Biotechnological Sciences, University of Catania, 95123 Catania, Italy; 5Endocannabinoid Research Group, Institute of Biomolecular Chemistry, Consiglio Nazionale delle Ricerche, 80078 Pozzuoli, Italy; 6Forensic Laboratory of Biologically Active Substances, Department of Chemistry of Natural Compounds, University of Chemistry and Technology Prague, 16628 Prague, Czech Republic; 7Psychedelic Research Centre, National Institute of Mental Health, Topolová 748, 25067 Klecany, Czech Republic; 8Central Nervous System Diseases Research (CNSDR), Boehringer Ingelheim Pharma GmbH & Co. KG, 88397 Biberach an der Riss, Germany; 9Canada Excellence Research Chair on the Microbiome-Endocannabinoidome Axis in Metabolic Health, Faculty of Medicine, Agricultural and Food Sciences, CRIUCPQ, INAF and Centre NUTRISS, Université Laval, Quebec City, QC G1V 4G5, Canada

**Keywords:** Δ^9^-tetrahydrocannabinol, methylazoxymethanol acetate, dopamine D2/D3 receptors, psychopathology

## Abstract

Adolescent exposure to cannabinoids as a postnatal environmental insult may increase the risk of psychosis in subjects exposed to perinatal insult, as suggested by the two-hit hypothesis of schizophrenia. Here, we hypothesized that peripubertal Δ^9^-tetrahydrocannabinol (aTHC) may affect the impact of prenatal methylazoxymethanol acetate (MAM) or perinatal THC (pTHC) exposure in adult rats. We found that MAM and pTHC-exposed rats, when compared to the control group (CNT), were characterized by adult phenotype relevant to schizophrenia, including social withdrawal and cognitive impairment, as revealed by social interaction test and novel object recognition test, respectively. At the molecular level, we observed an increase in cannabinoid CB1 receptor (*Cnr1*) and/or dopamine D2/D3 receptor (*Drd2, Drd3*) gene expression in the prefrontal cortex of adult MAM or pTHC-exposed rats, which we attributed to changes in DNA methylation at key regulatory gene regions. Interestingly, aTHC treatment significantly impaired social behavior, but not cognitive performance in CNT groups. In pTHC rats, aTHC did not exacerbate the altered phenotype nor dopaminergic signaling, while it reversed cognitive deficit in MAM rats by modulating *Drd2* and *Drd3* gene expression. In conclusion, our results suggest that the effects of peripubertal THC exposure may depend on individual differences related to dopaminergic neurotransmission.

## 1. Introduction

It is well accepted that exposure of the immature brain to several insults (such as prenatal infection, drugs of abuse, environmental pollutants, etc.) combined with genetic risk factors could lead to aberrant neurodevelopment, emerging psychopathology in adulthood including schizophrenia (SCZ), autism or obsessive–compulsive disorders [1,2,3,4,5]. *Cannabis* is the most common illicit drug used during pregnancy in Western countries, and its use is constantly increased in recent decades, irrespective of its detrimental health effects, which have long been documented [6]. Brain development begins in utero and extends through late adolescence [7], when it undergoes remodeling in the form of synaptic pruning, myelinization, and changes in neurotransmitter systems [8,9,10]. Indeed, prenatal age and adolescence are both crucial periods when exposure to drug abuse affects the central nervous system (CNS) development, leading to cognitive, emotional, and social alterations in adulthood, which are common hallmarks of psychopathology [11]. This is the cornerstone of the two-hit hypothesis of SCZ, stating that genetic and/or environmental factors might alter early CNS development producing long-term vulnerability to a “second hit” which would lead to the onset of SCZ symptoms. Therefore, gestational and/or adolescent exposure to Δ^9^-tetrahydrocannabinol (THC), the main psychoactive component of *Cannabis sativa,* might alter the normal activity of the endocannabinoid system (ECS) due to its high affinity for the cannabinoid CB1 receptor (*Cnr1*) gene [12,13], the overstimulation of which contribute to dysregulate the neurodevelopmental processes leading to neuroanatomical and functional alterations during neurogenesis [6,14]. A functional interaction between the dopaminergic (DAergic) system and the ECS due to the coexpression of *Cnr1* and dopamine D2 receptors (*Drd2*) genes [15] and to their signal transduction convergence [16] has been suggested both at the level of the prefrontal cortex (PFC) and in several limbic structures as possible neuroanatomical loci for neurodevelopmental disorders [11]. Thus, it appears to be likely that altered DAergic neurotransmission at the PFC level may also play a role in the long-lasting detrimental effects induced by perinatal insult per se or combined with peripubertal THC (aTHC) exposure. However, whether the potential effects of aTHC exposure as a “second hit” may be specifically related to the kind of pre/perinatal insult as a “first hit” is still debated.

Based on this background, the present study aimed to investigate if aTHC exposure could modify the effects of pre/perinatal insult induced by gestational methylazoxymethanol acetate (MAM) or perinatal THC (pTHC) exposure, two well-validated neurodevelopmental models of psychopathology [13,17,18], which are characterized by behavioral and molecular alterations similar to those described in subjects with neurodevelopmental disorders, further supporting their translational value [13,19]. More specifically, in adulthood, we assessed whether the cognitive deficits and social impairment typically described in the pTHC and MAM offspring and usually paralleled by changes in the ECS and/or DAergic system at the PFC level [13,17], might be modified by aTHC exposure (Figure 1).

## 2. Results

### 2.1. Peripubertal THC Exposure in Prenatally MAM-Exposed Rats

#### 2.1.1. Behavioral Effects

Prenatal MAM exposure alone or combined with aTHC treatment did not affect the spontaneous horizontal [number of crossings, factor MAM: F(1,45) = 3.379, *p* = 0.0727; factor aTHC: F(1,45) = 0.4158, *p* = 0.5223; MAM × aTHC interaction: F(1,45) = 0.4142, *p* = 0.5231] or vertical [number of rearings, factor MAM: F(1,45) = 2.657, *p* = 0.1101; factor aTHC: F(1,45) = 0.07309, *p* = 0.7881; and MAM × aTHC interaction: F(1,45) = 0.1337, *p* = 0.7163] locomotor activity in a novel environment (Figure 2A,B).

As shown in Figure 2C,D, prenatal MAM exposure impaired the social activity of adult offspring as revealed by the reduced time of interaction [*p* < 0.01 vs. CNT/aVEH group; two-way ANOVA, factor MAM: F(1,22) = 4.1999, *p* = 0.0526; factor aTHC: F(1,22) = 14.56, *p* = 0.0009; and factor MAM × aTHC interaction: F(1,22) = 8.410, *p* = 0.0083]. aTHC exposure alone or combined with prenatal MAM exposure affected social behavior as compared to CNT/aVEH group (*p* < 0.001; *p* < 0.01). No difference was detected in the number of interactions among the groups as an index of locomotor activity [two-way ANOVA, factor MAM: F(1,22) = 0.4513, *p* = 0.5087; factor aTHC: F(1,22) = 11.18, *p* = 0.0029; factor MAM × aTHC interaction: F(1,22) = 0.05968, *p* = 0.8093]. As depicted in Figure 2E, prenatal MAM exposure impaired the cognitive performance of adult rats, as described by reduced discrimination index in the novel object recognition test [two-way ANOVA, factor MAM: F(1,42) = 7.598, *p* = 0.0086; factor aTHC: F(1,42) = 1.261, *p* = 0.2679; factor MAM × aTHC interaction: F(1,42) = 14.17, *p* = 0.0005] as compared to CNT/aVEH group (*p* < 0.001). aTHC treatment reversed the cognitive deficit in MAM rats (*p* < 0.01 vs. MAM/aVEH), while it did not significantly affect the cognitive performance of CNT/aTHC (*p* > 0.05 vs. CNT/aVEH).

#### 2.1.2. Molecular Effects

Consistent with the increase in CB1 mRNA receptor expression [*p* < 0.05 vs. CNT/aVEH, *t*-test; two-way ANOVA, factor MAM: F(1,17) = 24.63, *p* = 0.0001; factor aTHC: F(1,17) = 0.4591, *p* = 0.5072; and factor MAM × aTHC interaction: F(1,17) = 0.2076, *p* = 0.6544, Figure 3B], we observed a significant reduction in DNA methylation of the CB1 (*Cnr1*) gene regulatory region in the four CpG average investigated in the PFC of MAM/aVEH group [*p* < 0.001 vs. CNT/aVEH; two-way ANOVA, factor MAM: F(1,17) = 31.10, *p* < 0.001; factor aTHC: F(1,17) = 37.90, *p* < 0.001; and factor MAM × aTHC interaction: F(1,17) = 19.68, *p* < 0.001, Figure 3A). Interestingly, in prenatally MAM-exposed rats, aTHC exposure reversed the DNA methylation of the CB1 (*Cnr1*) gene regulatory region (*p* < 0.001 vs. MAM/aVEH), but not the CB1 mRNA expression (*p* > 0.05 vs. MAM/aVEH). Prenatal MAM exposure also significantly upregulated the mRNA expression of dopamine D2 receptor in the PFC of adult rats [two-way ANOVA, MAM effect: F(1,15) = 7.950, *p* = 0.0129; aTHC effect: F(1,15) = 5.656, *p* = 0.0311; and MAM × aTHC interaction: F(1,15) = 11.18, *p* = 0.0044) as compared to CNT/aVEH group (*p* < 0.01), which was reversed by aTHC (*p* < 0.01, Figure 3D). Neither the prenatal MAM exposure nor the aTHC treatment affected the DNA methylation of the D2 (*Drd2*) gene regulatory region in the average of the six CpG sites investigated in the PFC of adult rats [two-way ANOVA, MAM effect: F(1,12) = 2.463, *p* = 0.1425; aTHC effect: F(1,12) = 1.709, *p* = 0.2156; MAM × aTHC interaction: F(1,12) = 0.6917, *p* = 0.4218, Figure 3C]. A significant inverse correlation between gene expression and discrimination index was observed for *Drd2* (Spearman r = −0.6978, *p* = 0.0101; Appendix A) but not for *Cnr1* (Spearman r = −0.1403, *p* = 0.6460; Appendix A). As depicted in Figure 3E, MAM insult increased the mRNA expression of dopamine D3 receptor in the PFC of adult rats [two-way ANOVA, MAM effect: F(1,16) = 5.871, *p* = 0.0281; aTHC effect: F(1,16) = 10.13, *p* = 0.0058; and MAM × aTHC interaction: F(1,16) = 4.557, *p* = 0.0486] as compared to CNT/aVEH group (*p* < 0.05), which was reversed by aTHC treatment (*p* < 0.01). aTHC failed to modify the dopamine D3 receptor expression in the CNT group (*p* > 0.05 vs. CNT/aVEH).

### 2.2. Peripubertal THC Exposure in Perinatally THC-Exposed Rats

#### 2.2.1. Behavioral Effects

Neither pTHC exposure nor aTHC treatment affected the spontaneous horizontal [number of crossings, two-way ANOVA, factor pTHC: F(1,38) = 0.1466, *p* = 0.7039; factor aTHC: F(1,38) = 0.03608, *p* = 0.5517; and pTHC × aTHC interaction: F(1,38) = 0.9920, *p* = 0.3255] or vertical [number of rearings, two-way ANOVA, factor pTHC: F(1,38) = 1.315, *p* = 0.2587; factor aTHC: F(1,38) = 0.6133, *p* = 0.4384; and pTHC × aTHC interaction: F(1,38) = 0.3950, *p* = 0.5334] locomotor activity in a novel environment (Figure 4A,B). The effect of aTHC treatment alone or combined with pTHC on behavioral performance in the social interaction test is shown in Figure 4C,D. Two-way ANOVA revealed for the time of interaction a main effect of pTHC [F(1,22) = 9.039, *p* = 0.0065], a main effect of aTHC [F(1,22) = 21.34, *p* = 0.0001), and a significant pTHC × aTHC interaction [F(1,22) = 8.989, *p* = 0.0066]. Post hoc analysis revealed that the pTHC/aVEH group spent less time in social interaction as compared to CNT/aVEH rats (*p* < 0.001), indicating impaired social behavior. aTHC affected the social performance in CNT/aTHC group (*p* < 0.001 vs. CNT/aVEH), but it did not modify it in the pTHC/aTHC group (*p* < 0.001 vs. CNT/aVEH). Neither pTHC exposure [F(1,22) = 0.6265, *p* = 0.4371] nor aTHC [F(1,22) = 4.258, *p* = 0.0511] affected the number of interactions [pTHC × aTHC interaction: F(1,22) = 0.02005, *p* = 0.8887], as an index of locomotor activity. In rats tested in the novel object recognition test, two-way ANOVA revealed a main effect of pTHC [F(1, 35) = 6.207, *p* = 0.0176), a significant pTHC × aTHC interaction [F(1,35) = 4.240, *p* = 0.0470], but not a main effect of aTHC [F(1, 35) = 3.142, *p* = 0.0850] for the discrimination index. Post hoc analysis revealed that pTHC exposure affected the recognition memory as described by the significant reduction in the discrimination index during the test phase (*p* < 0.05 vs. CNT/aVEH), which was not modified by aTHC treatment (*p* < 0.05 vs. CNT/aVEH). Furthermore, in the CNT group, aTHC did not significantly affect the cognitive performance of rats (*p* > 0.05 vs. CNT/aVEH; Figure 4E).

#### 2.2.2. Molecular Effects

Neither the pTHC exposure nor the aTHC treatment affected the DNA methylation of the CB1 (*Cnr1)* gene regulatory region in the average of the four CpG sites investigated in the PFC of adult rats [two-way ANOVA, pTHC effect: F(1,15) = 0.09692, *p* = 0.7598; aTHC effect: F(1,15) = 3.180, *p* = 0.0948; and pTHC × aTHC interaction F(1,15) = 0.003071, *p* = 0.9565; Figure 5A]. In the PFC of pTHC-exposed rats, there was a significant CB1 mRNA up-regulation [*p* < 0.01, vs. CNT/aVEH; two-way ANOVA, factor pTHC: F(1,15) = 6.901; *p* = 0.0190; factor aTHC: F(1,15) = 14.92, *p* = 0.0015; and pTHC × aTHC interaction: F(1,15) = 5.442, *p* = 0.0340] which was reversed by aTHC (*p* < 0.01). In the CNT group, the peripubertal exposure to THC did not modify CB1 mRNA expression (*p* > 0.05 vs. CNT/aVEH; Figure 5B).

The effect of aTHC exposure alone or combined with pTHC on mRNA expression of dopamine D2 receptors in PFC is depicted in Figure 5D. Two-way ANOVA revealed a significant pTHC × aTHC interaction [F(1,16) = 6.606, *p* = 0.0205], but neither a main effect of pTHC [F(1,16) = 3.848, *p* = 0.0674] nor a main effect of aTHC [F(1,16) = 0.8581, *p* = 0.3680]. Post hoc analysis revealed that pTHC exposure increased the dopamine D2 mRNA expression (*p* < 0.05 vs. CNT/aVEH), which was not significantly modified by aTHC (*p* > 0.05 vs. pTHC/aVEH). Furthermore, in the CNT group, aTHC did not significantly affect the D2 mRNA expression (*p* > 0.05 vs. CNT/aVEH). The pTHC exposure reduced the DNA methylation of the D2 (*Drd2*) gene regulatory region in the average of the six CpG sites investigated in the PFC of adult rats (*p* < 0.001 vs. CNT/aVEH, *t*-test), which was further reduced by aTHC (*p* < 0.01 vs. pTHC/aVEH, *t*-test). In the CNT group, aTHC did not significantly affect the DNA methylation of the D2 (*Drd2*) gene regulatory region [*p* > 0.05 vs. CNT/aVEH; two-way ANOVA, pTHC effect: F(1,13) = 28.09, *p* = 0.0001; aTHC effect: F(1,13) = 18.05, *p* = 0.0009; and pTHC × aTHC interaction: F(1,13) = 0.09454, *p* = 0.7634; Figure 5C). pTHC alone or combined with aTHC exposure failed to affect the mRNA expression of dopamine D3 receptor in the PFC of adult rats [two-way ANOVA, pTHC effect: F(1,16) = 4.268, *p* = 0.0554; aTHC effect: F(1,16) = 7.086, *p* = 0.0170; and pTHC × aTHC interaction: F(1,16) = 0.3213, *p* = 0.5787; Figure 5E].

## 3. Discussion

The findings of the present study confirm that pre/perinatal insult in rats evoked by maternal MAM or THC exposure causes harmful effects in offspring, namely (a) SCZ-like phenotype, (b) altered DAergic and/or cannabinoid neurotransmission, in a manner similar to that previously described both in preclinical studies and in SCZ subjects, thus further supporting the translational value of these two models of psychopathology [13,14,19,20,21,22]. More specifically, MAM or pTHC adult rats showed social and cognitive deficits, as described by the reduced time of interaction (as an index of social withdrawal) in the social interaction test and impaired short-term recognition memory in the novel object recognition test, which are often considered to be SCZ-like symptoms [11,23]. Furthermore, no difference was found in the number of interactions, as well as no spontaneous locomotor hyperactivity was described, in agreement with previous results [14]. The locomotor performance paradigms served as an internal control for possible unspecific stimulant effects; thus, our study reinforces the original findings that cognitive impairment in the novel object recognition test and social deficit in the social interaction test are robust phenotypes in pTHC or MAM rats, as valuable models of neurodevelopmental disorders.

MAM or pTHC exposure also induced alterations in the transcriptional regulation of the genes encoding for cannabinoid CB1 (*Cnr1*), dopamine D2 (*Drd2*) or D3 (*Drd3*) receptors at the level of PFC of adult rats, further confirming the hypothesis of DA–cannabinoid interaction as a molecular substrate of neurodevelopmental disorders [24]. However, MAM or pTHC exposure differentially affected the DNA methylation at *Cnr1* or *Drd2* gene regulatory regions, which were not always followed by significant alterations in gene expression. Thus, these findings suggest that alterations in dopamine D2 and cannabinoid CB1 mRNA levels evoked by pre/perinatal insults, which seem to be involved in the altered phenotype of adult rats, might also be regulated by other mechanisms at epigenetic levels, such as histone modifications, or most likely at post-transcriptional levels, e.g., via microRNAs [25]. Further investigations are indeed warranted to clarify these issues. We also observed a significant correlation between *Drd2*, but not *Cnr1* receptor gene expression and cognitive impairment in MAM rats in agreement with the hypothesis that altered DAergic neurotransmission at the PFC level may play a role in the cognitive deficits of SCZ as a neurodevelopmental disorder [26]. We cannot exclude that the hippocampus may also play a role in the cognitive performance of MAM or pTHC rats, since it is becoming increasingly clear that neurodevelopmental disorders are not only due to a circumscribed deficit in the PFC and/or hippocampus, but also represent a distributed impairment involving hippocampal–PFC connectivity [27]. However, in post-mortem studies, contradictory results have been found since decreased [23,28,29,30], increased [31], or unchanged [30,32] cannabinoid CB1 and/or dopamine D2 receptor brain expression has been detected. Conflicting data may be due to differences in patient symptom severity, amount of THC exposure, co-pharmacological treatments, or diagnostic methods in studies.

In addition to the well-documented detrimental effects of perinatal THC exposure, *Cannabis* use during adolescence is often associated with an increased risk of developing neuropsychiatric disorders in later life, due to changes at the level of GABAergic, glutamatergic, serotonergic, DAergic neurotransmission, and/or endocannabinoid signaling [6,33]. Evidence from animal studies consistently indicates that adolescent treatment with cannabinoid agonists induces lasting behavioral and molecular changes in adulthood, mimicking psychopathology [14]. In our experimental conditions, aTHC exposure significantly affected social activity, while it failed to induce a significant cognitive impairment in adult rats, partially in line with previous observations [6,14]. These discrepancies may be due to several factors, such as heterogeneity in *Cannabis* types (phytocannabinoid “THC” vs. synthetic cannabinoids “HU-210”, “CP-55940”, “WIN55,212-2”), doses, and timing of treatment (peripubertal vs. postpubertal), as well as to gender difference and to different cognitive tasks. At the PFC level, neither the cannabinoid CB1 nor the dopamine D2/D3 receptor gene expression was affected, suggesting that aTHC exposure may lead to social withdrawal by the involvement of different neurotransmitter systems, as previously suggested [14,33].

Based on the two-hit hypothesis, it has been suggested that SCZ, as a neurodevelopmental disorder, may be related to the combination of early prenatal and, likely environmental, postnatal insult. Consistently, combining exposure to prenatal insults and peripubertal stress in rodents may induce synergistic detrimental effects in adulthood [34]. Indeed, *Cannabis* exposure during adolescence which, per se, seems to be associated with an increased risk of developing psychopathology in adulthood [14], may induce more pronounced behavioral and molecular alterations in pre/perinatal stressed rodents. In pTHC rats, aTHC exposure did not further impair the social and cognitive deficits induced by pTHC exposure, suggesting that pTHC exposure at the dose of 5 mg/kg may induce strong detrimental effects, which cannot be further exacerbated by aTHC insult. At the molecular level, aTHC exposure significantly reversed the cannabinoid CB1 but not the dopamine D2 mRNA receptor overexpression induced by pTHC, suggesting a different impact of aTHC exposure on cannabinoid/DAergic system alteration induced by pTHC. Interestingly, aTHC exposure further reduced the dopamine D2 DNA methylation in pTHC, suggesting that THC exposure in adolescence may differentially impact epigenetic marks based on prenatal insult. As also mentioned above, we cannot exclude that another epigenetic mechanism, such as histone modifications and/or microRNAs, may also be involved in the effects elicited by aTHC exposure [25]. Overall, these findings indicate that aTHC exposure seems to solely revert the cannabinoid CB1 gene overexpression in the PFC of pTHC rats, whose reduction, in turn, does not impact the behavioral performances of pTHC/aTHC rats. Similarly, in MAM offspring, aTHC insult did not induce additive or synergistic detrimental effects at the level of social behavior. Although it may reflect a ceiling effect, the social withdrawal elicited by the aTHC exposure may be related to the involvement of different neurotransmitter systems, as recently suggested [14,25].

Contrary to our expectation, aTHC exposure improved the cognitive deficit of MAM rats in the novel object recognition test. How aTHC might reverse the cognitive impairment remains to be fully clarified. Based on the molecular findings, we assume that it may be due to the reduced dopamine D2/D3 receptor mRNA overexpression in the PFC of MAM rats, in agreement with previous studies showing that altered dopamine D2/D3 receptor expression may be involved in the cognitive deficits of neurodevelopmental disorders [13,21]. Notably, contrary to the pTHC model, aTHC did not revert the CB1 overexpression; thus, we could just speculate that in the neurodevelopmental MAM model, a pivotal role in the better cognitive performance induced by aTHC may be related to the changes in dopamine D2/D3 receptors more than the cannabinoid CB1 receptors. Further studies should clarify the involvement of different DAergic neurotransmission elements, such as dopamine D1-like receptors or dopamine transporter, both in the altered phenotype induced by pre/perinatal insults and in response to adolescent *Cannabis* exposure. To the best of our knowledge, this is the first study carried out in the MAM model of SCZ with peripubertal exposure to phytocannabinoid THC and not synthetic cannabinoids [35]. Specifically, adolescent exposure to the synthetic cannabinoid WIN55,212-2 attenuated the enhanced locomotor response to amphetamine in MAM rats, without impacting DA neuron activity at the level of the ventral tegmental area [35]. On the other hand, Aguilar et al., 2018 [36] showed that adolescent exposure to synthetic cannabinoids such as the CB1 agonist WIN55,212-2 or the FAAH inhibitor URB597 significantly increased the proportion of susceptible rats that developed an SCZ-like hyper-DAergic phenotype. Interestingly, our results are also in agreement with Lecca et al. (2019) [37], showing that adolescent THC intake attenuates the disruption of DAergic signaling in prenatal Poly I:C-exposed rats, a well-validated neurodevelopmental model of SCZ based on maternal immune activation [11]. The discrepancies with our results may be due to the different experimental designs, such as the pharmacology of the cannabinoids used (phytocannabinoid vs. synthetic; full vs. partial agonist; direct vs. indirect agonist), the adolescent cannabinoid protocol treatment (11 vs. 25 days), the generation of MAM offspring (first vs. second), and the brain region (PFC vs. ventral tegmental area).

In conclusion, our data confirm that different gestational insults such as MAM or THC exposure may lead at least to alteration of the DAergic (in terms of dopamine D2/D3 receptor expression) and endocannabinoid signaling (through the cannabinoid CB1 receptor altered expression), similarly to the changes previously described in SCZ subjects [13,19], further supporting the translational value of gestational MAM or pTHC exposure as neurodevelopmental animal models (Table 1). Although we just limited our study to assess the cannabinoid CB1 and dopamine D2/D3 receptor, we cannot exclude that the altered phenotype of MAM or pTHC rats, as neurodevelopmental animal models, may also be due to neuroimmune dysfunctions, as previously suggested [2,3,4,5]. On the contrary, the impact of aTHC exposure per se or combined with pre/perinatal insults is more complex than expected, and the combination of gestational and postnatal insults, as the main mechanism of the two-hit hypothesis of SCZ [34], needs to be further assessed. In this scenario, behavioral tasks for evaluating different cognitive and social aspects, as well as molecular analysis for assessing further potential neurochemical anomalies (i.e., immune cell dysfunction), should be helpful to better understand the effects of pre/perinatal insult alone or combined with postnatal insult.

## 4. Materials and Methods

### 4.1. Animals

Pregnant Sprague Dawley rats were obtained from Charles River (Germany) at gestational day (GD) 13 and housed individually. Environmental conditions during the whole study were constant: relative humidity 50–60%, temperature 23 °C ± 1 °C, normal 12 h light–dark cycle (light on 7 a.m. to 7 p.m.). Food and water were available ad libitum. All procedures were performed in accordance with EU Directive No. 2010/63/EU and approved by the Animal Care Committee of the Faculty of Medicine, Masaryk University, Czech Republic, and the Czech Governmental Animal Care Committee, in compliance with Czech Animal Protection Act No. 246/1992.

### 4.2. Drugs and Experimental Design

Methylazoxymethanol acetate (MAM; Midwest Research Institute, Kansas City, MO, USA) was dissolved in saline and administered intraperitoneally (i.p.) at dose of 22 mg/kg in 1 mL/kg volume on GD 17 with saline solution as a control (CNT), as previously described [17,18,38,39,40,41] (Figure 1, MAM model. THC (10 mg/mL in ethanol solution) obtained from the University of Chemistry and Technology, Prague, was prepared as previously described [42,43] (Figure 1, THC model). For perinatal administration, pregnant rats received daily oral gavage of THC (pTHC, 5 mg/kg/day) or sesame oil in control group (CNT) from GD 15 to postnatal day (PND) 9 [13]. The administered dose is equivalent to the current estimates of moderate exposure to THC in humans, correcting for differences in the route of administration and the body surface area [44]. No cross-fostering was used; the mothers were regularly weighed, and no differences in the body weight gains were observed among THC, MAM, or CNT-treated dams. The offspring were weaned on PND 22 and housed in groups of 2–3. For the peripubertal administration, THC was dissolved in vehicle solution (1% ethanol, 2% Tween 80, and saline). From PND 29 to PND 39, different groups of male CNT-, pTHC- or MAM-exposed rats (n = 12–15) were treated twice/day with increasing doses of THC (aTHC) or vehicle (aVEH), according to the following treatment schedule: 2.5 mg/kg/day (PND 29–31, i.p.), 5 mg/kg/day (PND 32–35, i.p.) and 10 mg/kg/day (PND 36–39, i.p.) as previously described [45]. Male rats were randomly assigned to the experimental procedures, and care was taken to avoid assigning more than two animals from the same litter to the same experimental group.

The drug treatment period in rats was carried out at the equivalent time of the childhood/periadolescent phase in humans [10], and it was also based on previous results both in MAM and pTHC [13,20] models. The adult (from PND 100) rats were subjected to behavioral tests with 5 days in between two consecutive tests, as previously described [46,47,48,49]. Immediately thereafter, they were decapitated in short ether anesthesia, and their brains were harvested. The PFC (corresponding to an area that included the rostral pole of the brain and delimited medially by the interhemispheric fissure, laterally by the corpus callosum, and caudally extended to AP + 2.7, according to Paxinos and Watson (1998) [50]) was dissected on ice by hand under microscopic control within 2 min, immediately frozen on liquid nitrogen, and stored at −80 °C until analysis.

### 4.3. Behavioral Testing

#### 4.3.1. Spontaneous Locomotor Activity in the Open Field Test (OFT)

The exploratory activity was assessed in moderately illuminated (80 lx) cubic metal arena (60 × 60 × 60 cm), as previously described [51,52,53,54,55]. Rats were placed gently in the center of the arena and allowed to explore. The vertical (number of rearing episodes) and the horizontal (number of squares crossed with all paws) exploratory activity was counted in 30 min sessions, recorded, and then scored offline by 2 observers blinded to the treatment groups. The arena was cleaned with 0.1% acetic acid and dried after each trial.

#### 4.3.2. Social Interaction (SI) Test

The test was carried out in a moderately illuminated room (120 lx), as previously described [17,43,46,47]. Each animal was allowed to freely explore an unfamiliar congener in a metal arena (60 × 60 × 60 cm) for 10 min. The arena was cleaned with 0.1% acetic acid and dried after each trial. Social behaviors were defined as sniffing, following, grooming, mounting, and nosing. The whole testing phase was recorded and analyzed by two observers blinded to the treatment groups. The time spent on social behaviors and the number of interactions were evaluated.

#### 4.3.3. Novel Object Recognition (NOR) Test

The experimental apparatus used for the NOR test was an arena (43 × 43 × 32 cm) made of plexiglass situated in a moderately illuminated room (120 lx). Each animal was placed in the arena and allowed to explore two identical, previously unseen objects for 5 min (familiarization phase). After an inter-trial interval of 3 min, one of the two familiar objects was replaced by a novel, previously unseen object, and rats were returned to the arena for the 5 min test phase [13,17,20]. Each phase was recorded and analyzed separately by two observers blinded to the treatment groups. The time spent exploring the familiar object (Tf) and the novel object (Tn) were scored. The discrimination index (DI) was calculated as DI = (Tn − Tf)/(Tn + Tf). The arena and all objects were cleaned with 0.1% acetic acid and dried after each trial.

### 4.4. Quantitative Real-Time Polymerase Chain Reaction (qRT-PCR)

Total RNA was isolated from rat dissected brain areas using TRIzol^®^ Reagent (Thermo Scientific, Waltham, MA, USA) according to the manufacturer’s protocol. The concentration of each purified RNA was detected by NanoDrop 2000c UV-Vis Spectrophotometer (Thermo Fisher Scientific, Waltham, MA, USA). The ratio of optical density at 260 and 280 nm was used to assess protein contamination: a value of 1.8–2.1 was considered acceptable. Starting with 0.5 µg of total RNA, complementary DNA (cDNA) was obtained with a mix of random hexamers and oligo-dT primers for mRNA using the RevertAid H Minus First Strand cDNA Synthesis Kit (Thermo Scientific, Waltham, MA, USA). Diluted cDNAs were then used to assess the relative abundance of each mRNA species by qRT-PCR, using the SensiFAST^TM^ SYBR^®^Lo-ROX Kit (Bioline, London, UK) on a 7500 Fast Real-Time PCR system (Thermo Fisher Scientific, Waltham, MA, USA). Each reaction mix was prepared with 2 µL of diluted cDNA, 7.5 µL of SensiFAST SYBR, and 10 pmol of each primer with a final volume of 15 µL. Sequences of the primers used for the amplification are reported in Appendix A. To provide an accurate quantification of the initial target in each PCR reaction, the amplification plot was examined, and the point of early log phase of product accumulation was defined by assigning a fluorescence threshold above background, defined as the threshold cycle number or Ct, as previously described [19]. Differences in threshold cycle number were used to quantify the relative amount of the PCR targets contained within each tube. After PCR, a dissociation curve (melting curve) was constructed in the range of 60 to 95 °C [56] to evaluate the specificity of the amplification products. The relative expression of different amplicons was calculated by the Delta–Delta Ct (ΔΔCT) method and converted to 2^−ΔΔCt^ for statistical analysis [57]. All data were normalized to two endogenous reference genes: glyceraldehyde-3-phosphate dehydrogenase (GAPDH) and beta-actin (*β*-ACT).

### 4.5. DNA Methylation Analysis by Pyrosequencing

Methylation status of *Cnr1* and *Drd2* genes was determined using pyrosequencing of bisulfite-converted DNA [13,19,20]. After extraction, DNA concentrations were estimated by measuring the absorbance at 260 nm, whereas sample purities were by considering the ratios of the absorbance values of 260 vs. 280 nm (A260/280 = 1.8). Each purified DNA was subjected to bisulfite modification by means of the EZ DNA Methylation-Gold^TM^ Kit (Zymo Research, Orange, CA, USA), according to the manufacturer’s protocol. DNA methylation status at each CpG site under study in the regulatory region of the genes (see Appendix A for gene details) was then assessed using a pyrosequencing assay. Bisulfite-treated DNA was first amplified by PyroMark PCR Kit (Qiagen, Hilden, Germany) with a biotin-labeled primer according to the manufacturer’s recommendations. PCR conditions were as follows: 95 °C for 15 min, followed by 45 cycles of 94 °C for 30 s, 56 °C for 30 s, 72 °C for 30 s, and, finally, 72 °C for 10 min. PCR products were then verified by agarose electrophoresis, immobilized to Streptavidin Sepharose High-Performance (GE Healthcare, Chicago, IL, USA) beads via biotin affinity, and denatured to allow the annealing with the sequencing primers. The sequencing was performed on a PyroMark Q24 ID using Pyro Mark Gold reagents (Qiagen). Rat *Cnr1* primers for PCR amplification and sequencing were generated according to PyroMark Assay Design software version 2.0 (Qiagen, Hilden, Germany) to analyze 4 CpG sites within the gene regulatory region. A specific PyroMark CpG assay (Qiagen, Hilden, Germany) was instead used to analyze the rat *Drd2* regulatory region, targeting 6 CpG sites. Details of the sequences under study and primers and assays employed are shown in Appendix A. Methylation’s level was analyzed using the PyroMark Q24 ID version 1.0.9 software, which calculates the methylation percentage mC/(mC + C) (mC = methylated cytosine, C = unmethylated cytosine) for each CpG site, allowing quantitative comparisons. Quantitative methylation results were expressed as the average methylation percentage of all the investigated CpG sites.

### 4.6. Statistical Analysis

The results are presented as group mean ± SEM. Behavioral data were first tested for normality distribution using the Shapiro–Wilk test. As normality tests have little power to detect non-Gaussian distributions with small data sets, we did not explicitly test for the normality of our biochemical data sets. Data were analyzed using two-way ANOVA (factor 1: model “MAM or pTHC”; factor 2: peripubertal THC treatment “aTHC”) followed by Tukey’s post hoc if appropriate. Unpaired *t*-test was used to analyze independent data. For correlation analysis, Spearman’s coefficient was used. Statistical evaluations were performed using specialized software (Graph-Pad Prism 9.0). A *p* level of 0.05 or less was considered indicative of a significant difference.

## Figures and Tables

**Figure 1 ijms-24-03907-f001:**
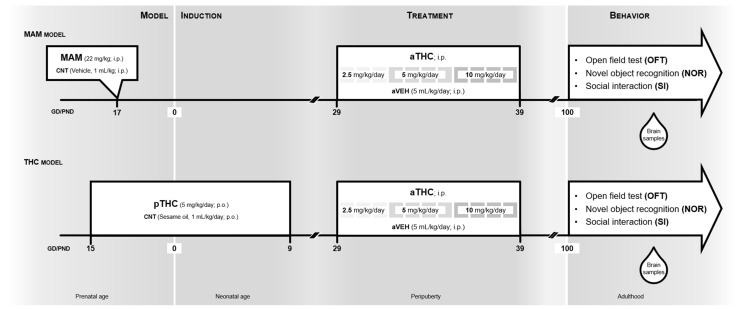
Timeline and experimental design used to investigate the effects of peripubertal THC (aTHC) exposure in two neurodevelopmental models of psychopathology. MAM model: pregnant rats were intraperitoneally (i.p.) treated with methylazoxymethanol (MAM) acetate (22 mg/kg) or saline (CNT: 1 mL/kg) on gestational day (GD) 17. From postnatal day (PND) 19 to PND 39, resulting male offspring were i.p. treated twice/day with increasing doses of THC (aTHC) or respective vehicle (aVEH). Behavioral tests (open field test, novel object recognition test and social interaction test) were conducted in adulthood from PND 100. After completion, the neurochemical analyses (quantitative real-time PCR and DNA methylation by pyrosequencing) were performed in the prefrontal cortex of rats. THC model: Pregnant rats were treated with THC (pTHC, 5 mg/kg/day; per os) or vehicle (CNT; 1 mL/kg/day; per os) from gestational day (GD) 15 to postnatal day (PND) 9. From PND 19 to PND 39, the resulting male offspring were i.p. treated twice/day with increasing doses of THC (aTHC) or respective vehicle (aVEH). Behavioral tests (open field test, novel object recognition test and social interaction test) were conducted in adulthood from PND 100. After completion, the neurochemical analyses (quantitative real-time PCR and DNA methylation by pyrosequencing) were performed in the prefrontal cortex of rats.

**Figure 2 ijms-24-03907-f002:**
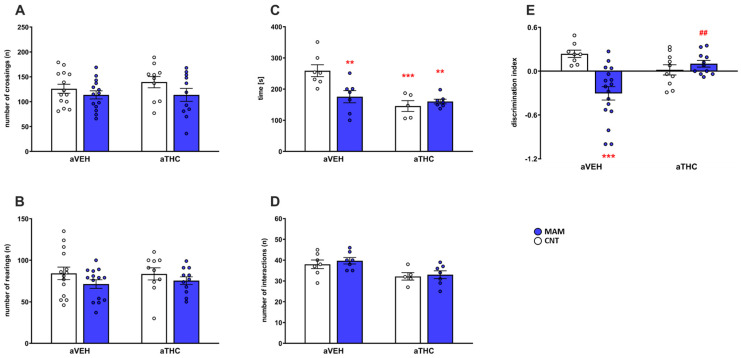
Effects of peripubertal treatment with THC (aTHC) on the behavioral phenotype of prenatally MAM-exposed rats in the open field test ((**A**): number of crossings; (**B**): number of rearings), in the social interaction test ((**C**): time of interaction; (**D**): number of interactions) and in the novel object recognition test ((**E**): discrimination index) in adulthood. Data are presented as means ± S.E.M. (n = 5–14 rats/group). ** *p <* 0.01 and *** *p <* 0.001 vs. CNT/aVEH; ## *p <* 0.01 vs. MAM/aVEH, Tukey’s post hoc test.

**Figure 3 ijms-24-03907-f003:**
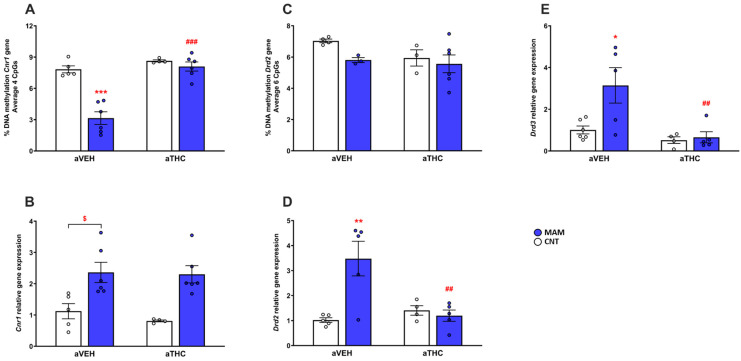
Effects of peripubertal treatment with THC (aTHC) on cannabinoid CB1 (*Cnr1*), dopamine D2 (*Drd2*) and D3 (*Drd3*) receptor genes in the prefrontal cortex of prenatally MAM-exposed rats in adulthood. Data are presented as means ± S.E.M. (n = 3–6 rats/group) of (**A**) DNA methylation of *Cnr1* gene (average 4 CpG sites), (**B**) *Cnr1* mRNA expression, (**C**) DNA methylation of *Drd2* gene (average 6 CpG sites), (**D**) *Drd2* mRNA expression and (**E**) *Drd3* mRNA expression. * *p <* 0.05, ** *p <* 0.01 and *** *p <* 0.001 vs. CNT/aVEH; ## *p <* 0.01 and ### *p <* 0.001 vs. MAM/aVEH, Tukey’s post hoc test; $ *p <* 0.05, unpaired-*t* test.

**Figure 4 ijms-24-03907-f004:**
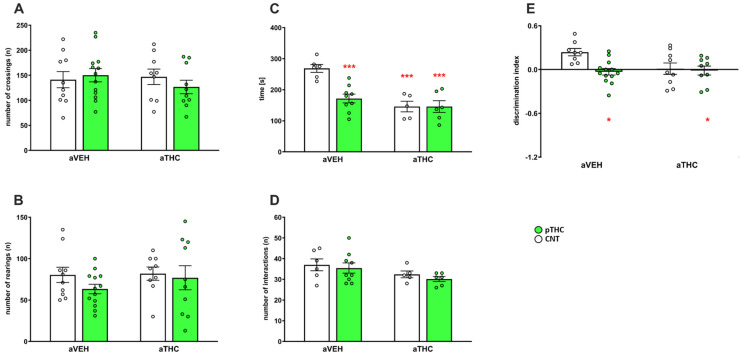
Effects of peripubertal treatment with THC (aTHC) on the behavioral phenotype of perinatally THC-exposed (pTHC) rats in the open field test ((**A**): number of crossings; (**B**): number of rearings), in the social interaction test ((**C**): time of interaction; (**D**): number of interactions) and in the novel object recognition test ((**E**): discrimination index) in adulthood. Data are presented as means ± S.E.M. (n = 5–13 rats/group). * *p <* 0.05 and *** *p <* 0.001 vs. CNT/aVEH, Tukey’s post hoc test.

**Figure 5 ijms-24-03907-f005:**
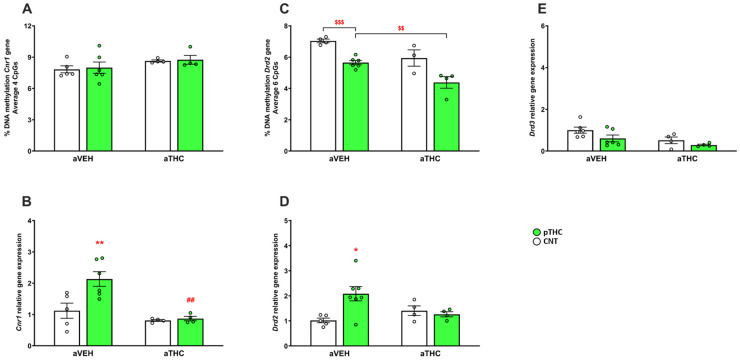
Effects of peripubertal treatment with THC (aTHC) on cannabinoid CB1 (*Cnr1*), dopamine D2 (*Drd2*), and D3 (*Drd3*) receptor genes in the prefrontal cortex of perinatally THC (pTHC)-exposed rats in adulthood. Data are presented as means ± S.E.M. (n = 3–8 rats/group) of (**A**) DNA methylation of *Cnr1* gene (average 4 CpG sites), (**B**) *Cnr1* mRNA expression, (**C**) DNA methylation of *Drd2* gene (average 6 CpG sites), (**D**) *Drd2* mRNA expression and (**E**) *Drd3* mRNA expression. * *p <* 0.05 and ** *p <* 0.01 vs. CNT/aVEH; ## *p <* 0.01 vs. pTHC/aVEH, Tukey’s post hoc test; $$ *p <* 0.01 and $$$ *p <* 0.001, unpaired-*t* test.

**Table 1 ijms-24-03907-t001:** Summary of the behavioral and molecular effects of peripubertal treatment with THC (aTHC) or vehicle (aVEH) in MAM or perinatally THC (pTHC)-exposed rats in adulthood.

Experimental Groups	Behavioral Effects			Molecular Effects (PFC)				
	OFT	SI	NOR	CB1 Gene	CB1 Meth	D2 Gene	D2 Meth	D3 Gene
CNT + aVEH	↔	↔	↔	↔	↔	↔	↔	↔
MAM + aVEH	↔	↓	↓	↑	↓	↑	↔	↑
MAM + aTHC	↔	↓	↑	↔	↑	↓	↔	↓
CNT + aTHC	↔	↓	↔	↔	↔	↔	↔	↔
pTHC + aVEH	↔	↓	↓	↑	↔	↑	↓	↔
pTHC + aTHC	↔	↓	↓	↓	↔	↔	↓	↔

↑, increase; ↓, decrease; ↔, no change.

## Data Availability

Data can be provided from the corresponding author upon reasonable request.

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
