# Peer review of "The Effects of Peripubertal THC Exposure in Neurodevelopmental Rat Models of Psychopathology"

_ijms, 2023, doi:10.3390/ijms24043907_

Round 1

Reviewer 1 Report

Comments for Authors

In this MSS entitled ‘Peripubertal THC exposure differentially affects molecular and 2 behavioral outcomes induced by neurodevelopmental insults’ by Martina Di Bartolomeo et al., studies described that Δ9-tetrahydrocan-32 nabinol (aTHC) may affect the impact of prenatal methylazoxymethanol acetate (MAM) or perinatal 33 THC (pTHC) exposure in adult rats.  This is well thought and designed study but poorly presented.

Some of the comments are as following.

Introduction is not written clearly.

One of the most striking things of this MSS is ‘Title’ which is not convincing and does not represent the work presented in this MSS. Second the choice of some words throughout the MSS is not proper.

Changes in Cnr1, Drd2, and/or Drd3 at mRNA levels are not sufficient enough in support of changes at molecular level.

Authors are advised to check the list of abbreviation and make sure the abbreviation is well elaborated at the of first use, example PND (reviewer is aware what PND stand for but may be not others).

Figure 1, looks incomplete, some key information are missing.

Text presented in line 90-99 is not clear. For example, authors wrote “male offspring were subjected to repeated i.p. treatment” what does it mean? It was repeated every 12 hrs or one day or two days.

Results section is very confusing and difficult to follow.

Results presented in Figure 2, 3 and 4 are not significantly changed except few cases.

Psychotropic ingredients in THC are known to induce changes like seen in schizophrenia in a dose dependent manner however data presented here partially fulfilled such hypothesis.

To study cognitive impairment, hippocampus is the brain region to study but authors focus on cortex.

Some of the strong statements made in this MSS are missing appropriate citation.

Too much content in discussion.  

Reviewer 2 Report

The study entitled Peripubertal THC exposure differentially affects molecular and behavioral outcomes induced by neurodevelopmental insults. Here are my comments:

1.     Abstract should be revised as it has several abbreviations which are difficult for the general reader to understand the aims of the manuscript. In the abstract section, the authors should write the background of the research. What motivated the researcher to

2.     Authors should provide more literature, I suggest the authors to elaborate introduction. The introduction needs to revision based on the update literatures (PMID: 31982447; PMID: 35259423; PMID: 30172736; PMID: 30790585).

3.     Additional information is needed on the justification for the animal numbers and how the animals were randomized?

4.     Which calculation method was used for RT-PCR analysis?

5.     Please write, n=? in figure legends and how many times the experiment was repeated.

6.     The discussion should start with a short paragraph containing the aims and major findings of the study. You should provide further evidence to support your hypotheses. Please discuss in light of previous findings.

7.     Please show individual datapoints of all figures.

8.     Conclusion: I recommend to the authors rewrite the conclusion section.

9.     The limitations of conducting their research are not shared in the discussion forum. The authors should present their study limitations, clarifying the scientific research community.

Round 2

Reviewer 1 Report

No comments required for authors

Reviewer 2 Report

The authors have answered all of my questions, and the paper has significantly improved.